Research

**Subject Area:**
developmental biology/systems biology

morphogenesis, growth modelling, plant development, volumetric growth, finite-element modelling

**Authors for correspondence:**
Richard Kennaway
e-mail: richard.kennaway@jic.ac.uk
Enrico Coen
e-mail: enrico.coen@jic.ac.uk

# Volumetric finite-element modelling of biological growth

Richard Kennaway and Enrico Coen

Cell and Developmental Biology, John Innes Centre, Norwich, UK

 RK, 0000-0001-9702-644X; EC, 0000-0001-8454-8767

Differential growth is the driver of tissue morphogenesis in plants, and also plays a fundamental role in animal development. Although the contributions of growth to shape change have been captured through modelling tissue sheets or isotropic volumes, a framework for modelling both isotropic and anisotropic volumetric growth in three dimensions over large changes in size and shape has been lacking. Here, we describe an approach based on finite-element modelling of continuous volumetric structures, and apply it to a range of forms and growth patterns, providing mathematical validation for examples that admit analytic solution. We show that a major difference between sheet and bulk tissues is that the growth of bulk tissue is more constrained, reducing the possibility of tissue conflict resolution through deformations such as buckling. Tissue sheets or cylinders may be generated from bulk shapes through anisotropic specified growth, oriented by a polarity field. A second polarity field, orthogonal to the first, allows sheets with varying lengths and widths to be generated, as illustrated by the wide range of leaf shapes observed in nature. The framework we describe thus provides a key tool for developing hypotheses for plant morphogenesis and is also applicable to other tissues that deform through differential growth or contraction.

## 1. Introduction

Various approaches have been taken to mathematically model plant tissue growth and morphogenesis in order to develop and clarify hypotheses for tissue behaviours [1–3]. Biologically, plant tissue is an interconnected mesh of cell walls, whose growth is driven by turgor pressure, with new walls arising by cell division as the tissue grows. One approach is to model the walls and interiors of cells explicitly. Such models allow cellular-level predictions to be made and validated. This method has been used to model the growth of meristem primordia, embryos, roots, shoot apices, fern gametophytes, sepals and leaves [4–17].

Another approach, and the one adopted in this paper, is to abstract from cells, by modelling tissue as a continuous material, either as a surface with no thickness or as a sheet of finite thickness. The advantage of the continuum approach is its simplicity, particularly when dealing with three-dimensional deformations and anisotropic growth. The method has been used to simulate growth and morphogenesis of various tissue sheets, including shoot apices, leaves, sepals, petals and fruits [2,4,18–27], as well as animal tissue sheets such as hearts and facial cartilage [23,28].

Modelling software that we have developed [18] was specialized for sheet-like tissues (those that are everywhere thin in one direction, but may be curved in space). Here, we extend the method and software to volumetric growth of bulk tissues. Sheet-like tissues can then be modelled as an outcome of patterns of growth from an initial non-sheet-like primordium, rather than being an assumed initial state.

When tissue is modelled as a continuum, it is assumed that for each region of the tissue, there is a specified rate of growth that defines how much that

region would grow in mechanical isolation from neighbouring tissue [18]. This rate of 'specified' growth is a tensor quantity, representing the possibility that the growth may be by different amounts in different directions. The problem is then to compute the deformation field, or resultant growth (i.e. a mapping of each region of the tissue to its new position), that will result from applying the field of specified growth rate for all regions when mechanically connected together, over some small time interval. In general, there will be no deformation field in which every region of the tissue achieves its specified growth. For example, a disc-shaped tissue growing faster at its edge than at its centre may be forced to buckle out of the plane, introducing a deformation (curvature or rotation) which was not specified by the growth field. The actual deformation resulting from the field of specified growth is taken to be whatever shape minimizes the strain energy. In general, there may be no deformation that reduces the strain energy to zero everywhere, and there will remain some residual strain. Another way of conceptualizing this is that the specified growth defines a new metric on the tissue, and the deformation is that which aligns this metric as closely as possible (i.e. with minimal strain energy) with the standard Euclidean metric [29]. Residual strain may be assumed to dissipate as fast as it arises, or may be allowed to accumulate at a finite rate.

The physical mechanism of growth of plant tissue is generally held to be that cells enlarge by turgor pressure acting to stretch cell walls whose elastic stiffness has been reduced, which are then reinforced by the addition of wall material [30]. In our framework, the specified growth of each cell-sized region during a time step corresponds to the amount by which the cell would incrementally yield in this way, if it were unconstrained by the surrounding tissue. The resultant growth of each region is then its equilibrium conformation given the constraint that the whole tissue remains continuous.

The discarding of residual strain after each time step amounts to treating the resultant deformation as an irreversible plastic flow. Thus, the framework treats growth as a viscoelastic process [31]. The amount of plasticity may be reduced by retaining all or part of the residual strain.

Specified growth rates are assumed to be set by morphogens that are produced or removed at certain locations, and may diffuse through the tissue and react with each other. These processes are modelled by partial differential equations, which are solved by numerical methods. The sheets are one finite element thick, and growth is specified as being oriented normal to the plane of the sheet (growth in thickness) or oriented within the plane (planar growth). Anisotropic growth within the plane can be specified through a polarity field, established by taking the local gradient of a diffusible factor called POL. Specified growth rates may then be given parallel ($k_{par}$) or perpendicular ($k_{per}$) to the gradient of POL. Sheet curvature may arise as it reduces potential growth conflicts [1,32].

To extend this framework to volumetric growth, several challenges need to be met. First, methods are required for dynamic local subdivision of finite elements growing in three dimensions, in order to maintain the computational quality of the mesh as it is deformed. Second, anisotropic growth orientations need to be specified through an internal mechanism, rather than by pre-assigning growth to be within or normal to the plane of a sheet. Third, growth patterns should be validated by examples that allow analytical solutions. Here, we address these problems and apply the volumetric framework to several case studies, illustrating how additional tissue constraints arise in bulk tissue compared with sheets, and how a variety of forms, including sheets and outgrowths, can be generated.

## 2. Results

### 2.1. Volumetric finite-element meshes

Modelling the growth of two-dimensional sheets is in some ways simpler than modelling growth in three-dimensional volumes, because growth orientations are specified in relation to the plane of the sheet (i.e. growth parallel to the plane and growth in sheet thickness). This difference influences the choice of finite-element shapes. For sheets, we employed finite elements in the shape of warped triangular prisms [18], illustrated in figure 1. Sheets are one element thick (modelling sheet thickness directly rather than by thin shell methods), with the triangular ends of the prisms corresponding to the two surfaces of the sheet.

For solids of arbitrary shape, tetrahedral elements are more convenient. Any shape can be decomposed into tetrahedra (figure 1 shows an example), whereas other finite-element shapes are less flexible. A further advantage of tetrahedral elements is that they are more amenable to local dynamic remeshing (discussed below). All of the solid three-dimensional models described in this paper use meshes of tetrahedra.

For simplicity of coding, we have implemented only first-order (linear) elements, as employed previously for modelling sheets [18]. That is, the geometry of an element is determined by its corner vertices, and there are no additional vertices along its edges, within its faces or in its interior. Each quantity of interest, such as the concentration of a morphogen, is represented by its values at the vertices of the elements, and conceptually interpolated over the volume of each element. Each step of the simulation calculates the specified deformation of each finite element at certain points in its interior, and the result of the calculation is a displacement of every vertex. This displacement is then applied, and a new simulation step begun.

The constitutive equations, which define the relevant physical properties of the tissue (elasticity and diffusion), are the same as those previously used (see [18] and the electronic supplementary material to the present paper). The numerical implementation applies them to general tetrahedral meshes instead of a single layer of pentahedra.

### 2.2. Dynamic refinement

Differential regional growth can lead to the accumulation of very large differences in element size over a mesh. Anisotropic growth can also produce elements that are very thin in one or two dimensions. These phenomena can reduce numerical accuracy. To maintain the quality of the decomposition into finite elements, it is necessary from time to time to subdivide or reorganize parts of the mesh, by splitting large elements into smaller ones and eliminating very thin elements, while maintaining the outer shape, the continuity of the mesh and the distribution of morphogens across the mesh.

royalsocietypublishing.org/journal/rsob Open Biol. 9: 190057

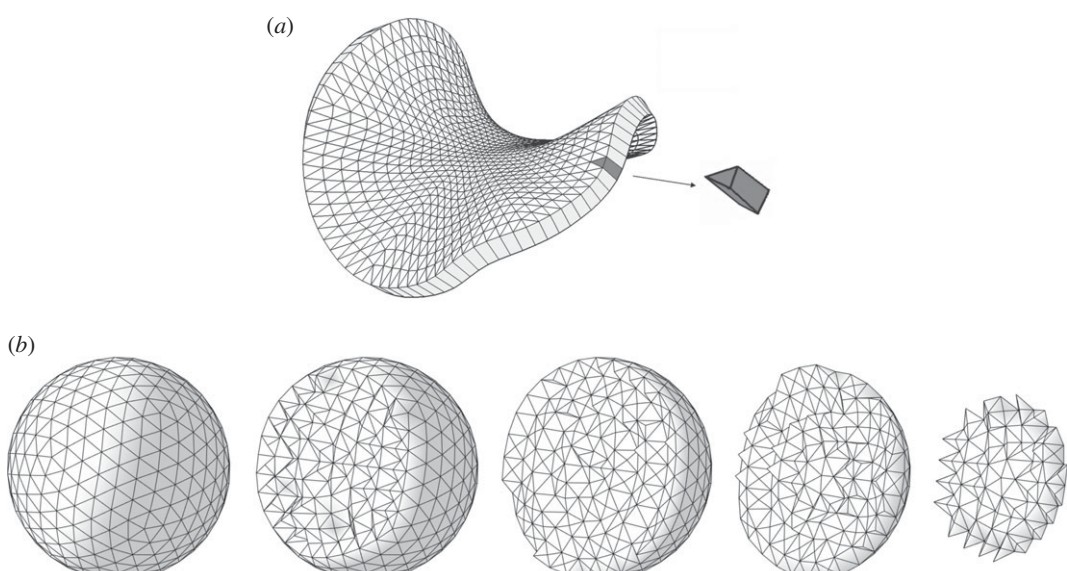

**Figure 1.** (*a*) A curved sheet divided into 1536 warped triangular prisms (image reproduced from [18]). (*b*) Successive cutaway views of a solid sphere divided into 7120 tetrahedra.

This process of dynamic remeshing involves splitting finite elements in such a way that all daughter elements are of the same type as the parents (e.g. all tetrahedra, or all prisms). Dynamic remeshing of a growing sheet of triangular prisms is relatively straightforward, as we only need to consider subdividing the triangular surface mesh (there are no subdivisions within the thickness of the sheet). If we wish to split a single edge of a triangle by introducing a new vertex (figure 2*a*), we can simply split the triangle on each side into two. Similarly, splitting two or three edges with new vertices (red in figure 2*b* and *c,* respectively) can create daughter triangles by joining to surrounding vertices. If an arbitrary set of edges is to be split, we can find for each triangle a subdivision into smaller triangles that splits exactly the desired edges (figure 2*d*).

Dynamic subdivision of a tetrahedral mesh involves establishing new connectivities with vertices in three dimensions. For example, figure 3*a* shows a mesh comprising five tetrahedra around a common edge (drawn in red). The individual tetrahedra are shown in an 'exploded view' in figure 3*b*. If we split the common edge by introducing a new vertex (also shown in figure 3*a*), then all five of the tetrahedra must be split in two as illustrated in figure 3*c*. The 10 tetrahedra are shown in exploded view in figure 3*d*. However, if an arbitrary subset of the edges of a tetrahedron are to be split, then it becomes more complicated. When a face of a tetrahedron is split, and that face is shared by a second tetrahedron, then that tetrahedron must be split in a way that gives the same splitting of the common face in order to maintain consistency of the finite-element decomposition. There are further transformations, such as merging the ends of very short edges, that would be useful to maintain the quality of the mesh but implementing these is not straightforward (see electronic supplementary material).

We implemented dynamic subdivision of the tetrahedral mesh for the following cases.

(1) Only one edge is to be split (figure 4*a*).
(2) A pair of opposite edges is to be split (figure 4*b*).
(3) The three edges of one face are to be split (figure 4*c*).
(4) Every edge is to be split (figure 4*d*).

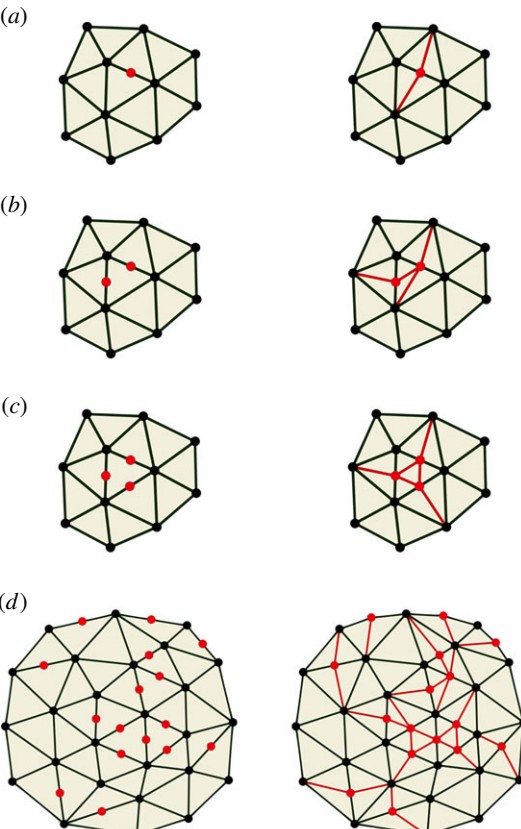

**Figure 2.** Dynamic subdivision of a triangular mesh. Splitting one (*a*), two (*b*) or three (*c*) edges of a triangle in a surrounding mesh, and splitting an arbitrary subset of the edges in a mesh (*d*). New vertices are drawn in red on the left and new edges added to maintain a triangular mesh are drawn in red on the right.

We do not implement cases where at least one face of a tetrahedron contains exactly two of the edges to be split, as in the front face of the example in figure 4*e*. The reason is that, while in cases (*a–d*) there is a natural choice of how to subdivide each face of the tetrahedron, in cases such as (*e*) there are two different ways to split the face. Another tetrahedron will share that face and must give it the same subdivision. Coordinating these divisions while searching

**4**

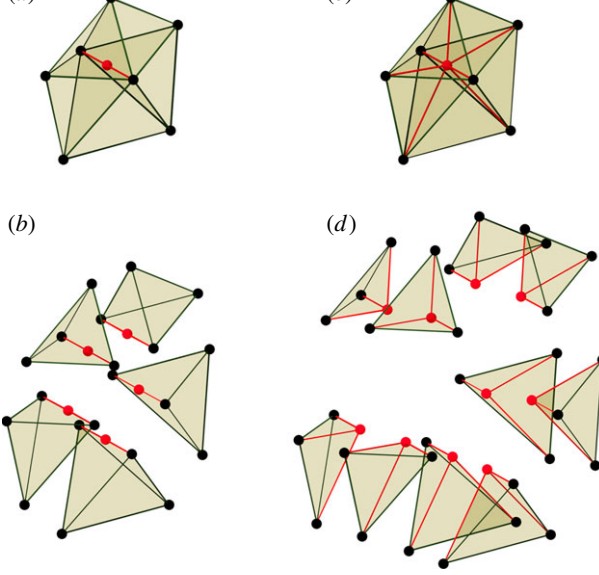

**Figure 3.** Dynamic subdivision of a mesh of tetrahedra. (*a*) Initial mesh of five tetrahedra around a common edge (shown in red) to be split with a new vertex. (*b*) Exploded view of the initial mesh. (*c*) Result of splitting. (*d*) Exploded view of the result.

for one that will best maintain the quality of the mesh is a complex problem.

To ensure that we only deal with cases (*a*–*d*), when we have identified a set of edges that should be split because of their length, we eliminate edges from the set until the required splitting of each tetrahedron matches one of those cases. The edges left unsplit can be dealt with by repeated applications of this process.

For meshes made of pentahedra (warped triangular prisms) and hexahedra (warped cubes), dynamic remeshing is more difficult. It is easy to subdivide every element simultaneously into smaller elements, but to refine some arbitrary subset, while maintaining consistency across shared faces, is not necessarily possible, unless one allows meshes that mix different shapes of element. The number of special cases of thin elements of these shapes is also large. Therefore, while our software supports pentahedral and hexahedral elements, we have not implemented remeshing for these elements.

As described in [18], when we subdivide sheet meshes, new vertices are in general not placed at the midpoints of the split edges, because this would tend to preserve the flatness of every triangle (figure 5*a*). Instead, we use a method drawn from computer graphics called butterfly subdivision [33] (figure 5*b*). This method considers the triangular mesh to be an approximation to a continuous curved surface, and attempts to place the new vertex on that surface by suitably offsetting it from the midpoint. For volumetric meshes, we use simple bisection for edges in the interior, but butterfly subdivision on the surface.

Many subdivision methods exist in the graphics literature [34]. For our purposes, butterfly subdivision has the essential property that the refined mesh includes all of the vertices of the original mesh. This is not the case, for example, for the more widely used Catmull−Clark subdivision [35], where none of the original vertices lie on the refined mesh, and which tends to shrink the mesh as it is refined (figure 5*c*). The Catmull−Clark method also produces quadrilaterals, regardless of what polygons the original mesh was made of.

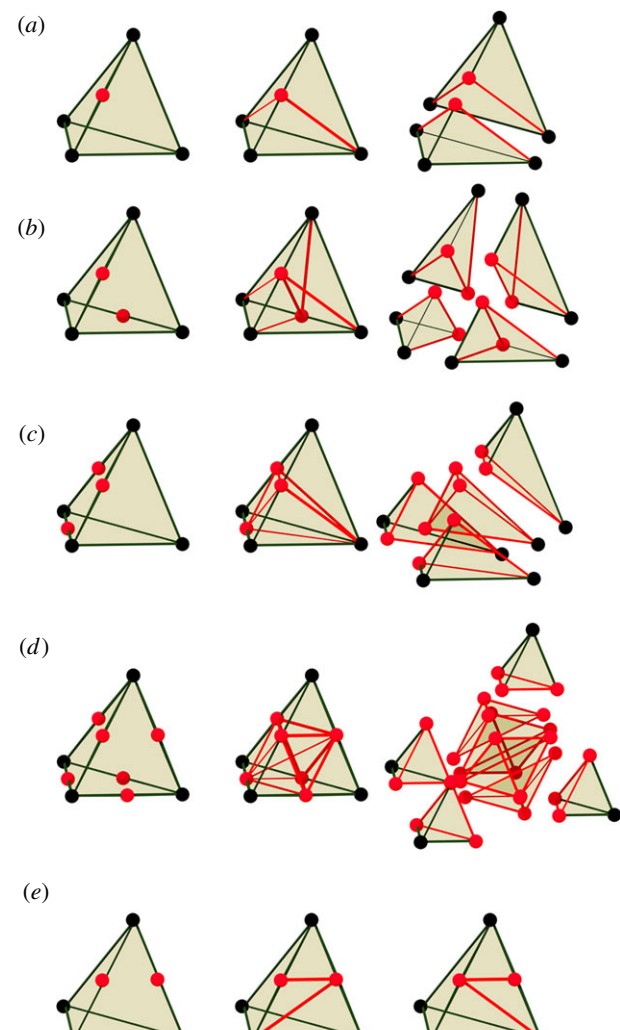

**Figure 4.** Implemented cases of subdivision. (*a*–*d*) Left column: initial tetrahedron with newly introduced vertices (red). Middle column: division into tetrahedra. Right: exploded view. (*a*) Only one edge is to be split. (*b*) A pair of opposite edges are to be split. (*c*) Three edges of one face are to be split. (*d*) Every edge is to be split. A tetrahedron is sliced off each corner of the original, leaving an octahedron which is then sliced into four more tetrahedra around one of its diagonals. (*e*) Example of a case that is not implemented. Two adjacent edges are to be split. There are two ways to split the face containing these edges into triangles. The tetrahedron on the other side of that face must split it in the same way. This coordination of splitting complicates the implementation.

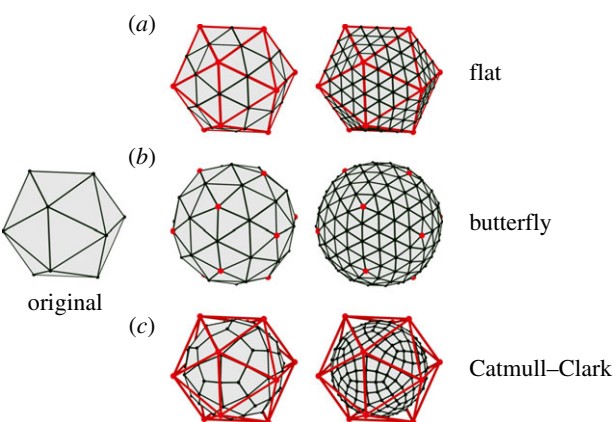

**Figure 5.** Two iterations of three methods of subdivision: flat (*a*), butterfly (*b*) and Catmull−Clark (*c*). The original mesh is superimposed in red on the subdivided meshes.

royalsocietypublishing.org/journal/rsob    Open Biol. **9**: 190057

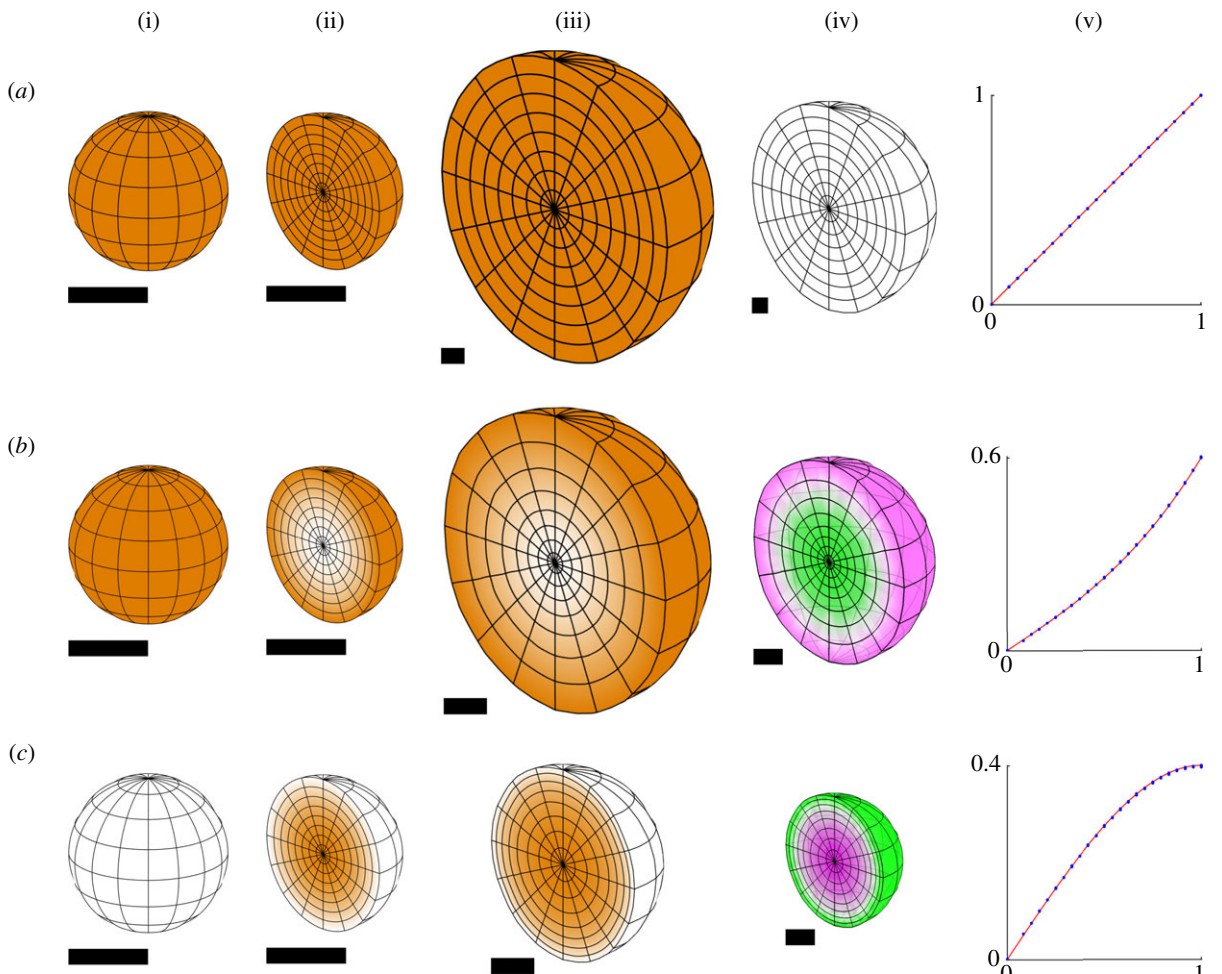

**Figure 6.** Isotropic specified growth in a sphere of unit radius (with respect to arbitrary units of space and time). (*a*) Uniform growth rate, (*b*) growth rate is $r^2$ and (*c*) growth rate is $1 - r^2$. Each row shows (i) the initial distribution of growth, (ii) the same in a hemispherical cutaway view, (iii) the growth distribution in the final state, (iv) the total residual strain in the final state and (v) the initial radial displacement rate plotted against the initial radius for the finite-element solution (blue dots) and an analytical solution (red line). Each dot represents a large number of vertices at the same initial radius, 12 245 vertices in total. Orange represents specified growth rate, and magenta/green is the rate of generation of residual compression/tension. Scale bars are equal to the initial radius of the sphere, taken to be 1 unit. The duration of growth is chosen to make the general pattern clear.

The displacement of vertices prescribed by the butterfly method should not be thought of as a deformation of the finite elements, but as a remeshing that better approximates the smooth-surfaced volume that is being modelled. Therefore, if residual strain is present in the elements, it retains its values during subdivision. The volume of a divided element at the surface will increase slightly where the surface is convex and decrease where it is concave, thus changing its total strain energy, but the amount is slight and comparable to the finite accuracy of the finite-element method itself.

## 2.3. Volumetric tissue conflicts

Using the finite-element system described above, we explore the effects of various spatial patterns of specified growth on the resultant growth and deformation. For growth of sheets, tissue conflicts can be classified into three types: surface (differential specified growth between the surfaces of the sheet), areal (differential specified growth rates across the sheet) and directional (variation in the orientation of specified growth) [32]. For volumetric growth modelling, there is no predefined sheet, so there is no category of surface conflict. We, therefore, define only two types of conflict: regional

(conflicts arising through variation in specified volumetric growth rates) and directional (conflicts arising from variation in the orientations of specified growth). We investigate how both regional and directional tissue conflicts may be resolved and how this compares with the situation for growth of sheets.

## 2.4. Isotropic specified growth

We first consider cases in which specified growth is equal in all orientations. In this case, specified growth rates for each region of tissue can be described with a scalar: the specified volumetric growth rate. Because of tissue conflicts, resultant growth rates may include anisotropic and rotational (vorticity) terms, and therefore require tensors to be fully represented.

### 2.4.1. Uniform isotropic growth

The simplest type of growth is uniform specified isotropic growth, which produces an enlargement of the tissue, whatever its initial shape with no conflict or residual strain. This is illustrated in figure 6*a* for a sphere. A latitude/longitude grid is shown on the outside of the initial spherical tissue,

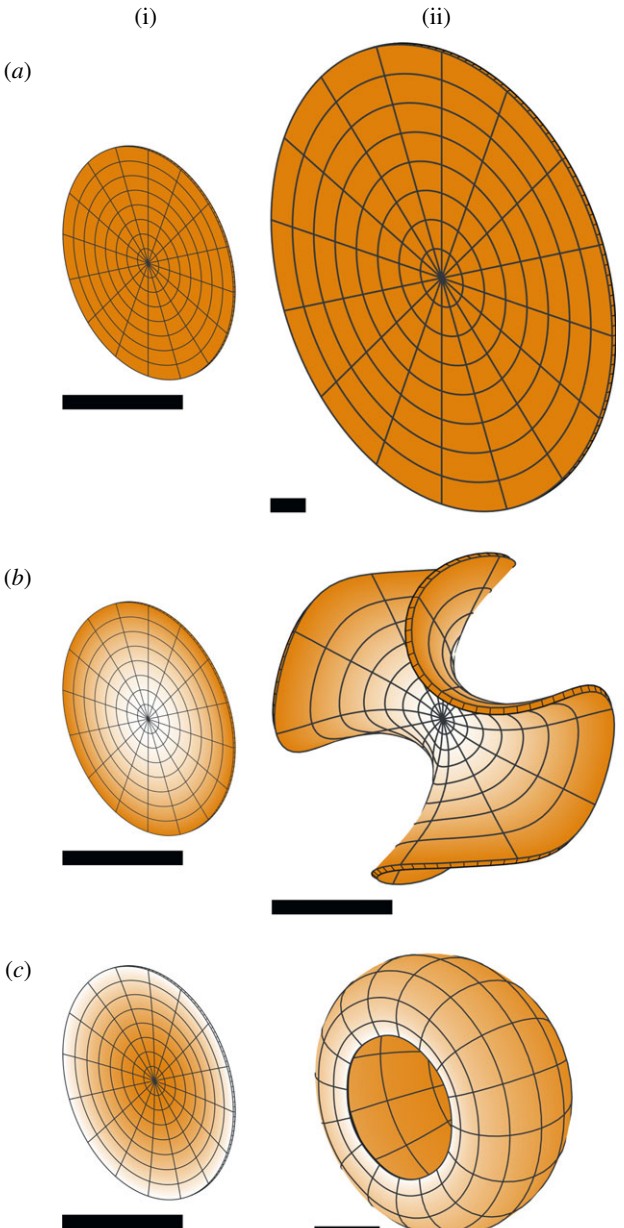

**Figure 7.** Isotropic specified growth in a disc of unit radius (shown obliquely), with the same three growth distributions as figure 6: (*a*) uniform, (*b*) $r^2$, (*c*) $1 - r^2$. Each row shows (i) the initial distribution of growth (orange) and (ii) the growth distribution in the final state. The finite-element decomposition of the disc consists of a single layer of pentahedra (prisms), obtained by dividing a planar disc into triangles and giving them an initially uniform thickness. Specified growth is still isotropic in all three dimensions. A small initial perturbation (invisibly small in the picture of the initial state) is given to each vertex of the finite-element mesh, to break the symmetry and allow buckling to occur. Orange represents specified growth rate. Scale bars are equal to the initial radius of the disc.

and a section shown in a hemispherical cutaway. As expected, the shape is unchanged after growth, there is no residual strain and there is a good agreement with the radial displacement obtained with the analytical solution. Similar results are obtained with an initially circular sheet (figure 7*a*).

In this and in all of the following figures, units of space and time are arbitrary. The initial shape is always a sphere or disc of unit radius, or a cube or square of unit semi-diameter. Growth rates have dimension 1/time. With respect to the arbitrary time unit, maximum growth rates in the

examples are equal to 1. The simulations are computed with a time step of 0.1 or less, so that the amount of specified growth in any direction over a single time step is never more than 10%. The duration of each simulation varies and is chosen to make the general form of the development clear. In images with scale bars, all scale bars have unit length. In figures without, all images are to the same scale.

### 2.4.2. Isotropic growth varying along the radial axis

For many plant tissues, the outer epidermal layer is under residual tension and the inner under residual compression, suggesting differential specified growth along the inside–outside axis [36]. We explore this pattern of growth by having the rate of specified isotropic growth in our sphere depend on the radial distance from the centre, creating regional conflicts.

In cellular terms, the notion of residual compression does not mean that plant cell walls themselves are under compression, only that the region of tissue is held at a smaller size than it would otherwise take without the external constraint. Thus cell walls will still be under tension owing to osmotic pressure, even in the presence of residual compression.

If the specified growth rate of the sphere increases with radial distance (proportional to the square of the radius), the shape remains spherical, with the inner core regions experiencing residual tension, while the outer surface is put under residual compression (figure 6*b*). The lack of shape change contrasts with what happens if a circular sheet rather than a sphere has the same pattern of specified growth. For a disc, the outer rim of the sheet buckles (figure 7*b*), resolving most of the residual strain. The number and positioning of the ripples depends on the rapidity of the increase of growth with radius, the initial perturbation and the thickness of the sheet. (Since the initial perturbation is random for (*b*), the buckling is different each time this simulation is run.) Thus, compared with a sheet, the constraint of the surrounding material in a solid sphere inhibits regional conflict resolution through buckling. While buckling can occur in a solid sphere ([29], ch 15), it requires more extreme disparities in growth or elasticity properties than it does in a thin sheet, where buckling will result from even quite small amounts of residual strain.

If the specified growth rate of the sphere decreases with radial distance (decreasing quadratically from a maximum at the centre to zero at the surface), the shape again remains spherical, but now the core regions experience residual compression, while the outer surface is put under residual tension (figure 6*c*), similar to the situation for some plant tissues. With the same pattern of specified growth a circular sheet buckles to form a cup shape (figure 7*c*), resolving most of the residual strain except for some stretching near the rim. These results illustrate how volumetric shapes are less able to resolve regional conflicts through changes in curvature than sheets, and thus have more constrained shapes.

The symmetry of these examples allows for an alternative mathematical analysis, taking advantage of the symmetry to reduce the problem to an ordinary differential equation. When the growth rate is a sum of powers of the radius (including non-integer powers), this equation can be solved explicitly (see electronic supplementary material). This provides a check on the correctness of the finite-element implementation. Figure 6 includes graphs comparing the finite-element

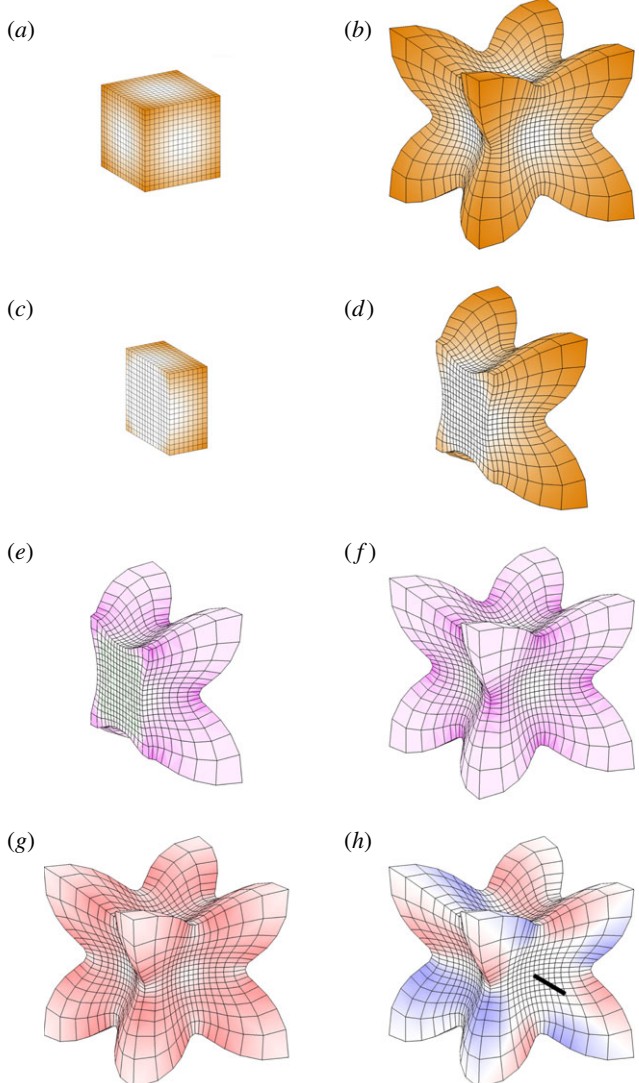

**Figure 8.** Non-uniform isotropic growth in a cube. Specified growth is zero from the centre out to the face centres, thereafter increasing linearly with radius to a maximum at the corners. All images are to the same scale. (*a–d*) Initial and final tissue with specified growth rate in orange, shown as the entire volume (*a,b*) or section (*c,d*). (*e,f*) Residual strain shown as a section (*e*) or entire volume (*f*). Magenta indicates compression; green tension (almost invisibly low, but present in the central region). (*g,h*) Vorticity. The magnitude of the rotation rate (*g*) is indicated by the intensity of red. (*h*) Rotation rate about one of the coordinate axes (indicated by the black line), with red representing anticlockwise rotation when viewed looking along that axis, and blue clockwise.

calculations with the analytical solutions. The displacement rate is plotted against the radius in the initial state. The analytical solutions are respectively $r$, $0.4r + 0.2r^3$ and $0.6r - 0.2r^3$.

The sphere shows no overall change of shape in these examples (it remains a sphere), owing to the common spherical symmetry of the tissue and the growth pattern. Radial growth patterns can produce shape change in other volumetric shapes, such as cubes. Figure 8*a–d* illustrates the result of a growth rate applied to a cube, which is zero everywhere from the centre of the cube out to the distance of the face centres, thereafter increasing linearly with radius to a maximum at the corners. Unlike the sphere, there is a shape change, and protrusions form at the corners. There is reduced residual compression towards the corners as these have the most freedom to deform into the surrounding

space (figure 8*e,f*). A further difference from the sphere is presence of vorticity. For spheres with a radial gradient of growth, there was no rotation of the deforming tissue (zero vorticity everywhere). Each point moves only in the radial direction. However, vorticity is generated for the cube (figure 8*g*). The direction of rotation with respect to a particular axis is shown in figure 8*h*. The vorticity reflects changes in curvature, which reduce residual strain. The opposite pattern of specified growth rate reducing with distance from the centre of a cube would tend to stretch and flatten the corners. These comparisons between a sphere and a cube illustrate how introducing asymmetries (e.g. corners with more space around them to grow into) gives greater freedom for curvature and changes in shape to arise through regional conflicts.

### 2.4.3. Isotropic growth varying along a linear gradient

If a linear gradient of isotropic specified growth is set up in a cube (figure 9*a*), and is thereafter fixed in the tissue and moves with it, the cube deforms (figure 9*b*). The resultant growth is not locally isotropic. The transformation in shape exhibits both vorticity (figure 9*e*) and some residual strain (figure 9*f*). The same specified growth pattern with a square sheet yields a locally isotropic (conformal) mapping (figure 9*c,d*). In this case there is still vorticity (figure 9*g*) but negligible residual strain (figure 9*h*), showing that potential regional growth conflicts have been largely resolved. Thus, compared with the growth of sheets, the growth of volumetric shapes provides additional constraints, making it more difficult to fully resolve regional conflicts. This is also evident from the Riemann mapping theorem, which shows that in two dimensions isotropic mappings exist between almost any two shapes that do not have holes. In three dimensions, there are by comparison very few such mappings.

If a linear gradient of isotropic growth is incorporated in a sphere, rather than a cube, a flattening or indentation at the base is generated (figure 10*a,b*). The residual strain pattern shows the indentation to be under residual tension while the sides are under residual compression (figure 10*c*). The shape change depends on the nature of the growth gradient. If a gradient in specified growth rate is set up from a source at the 'north pole' with diffusion and decay, an egg shape is generated (figure 10*d–f*).

## 2.5. Anisotropic specified growth along one field of orientations

We now consider the forms of growth that can arise when specified growth is anisotropic. In this section, we consider anisotropic growth oriented by one polarity field established through diffusion of a morphogen termed POL.

### 2.5.1. Polarity field

To model anisotropic specified growth of sheets, three orthogonal axialities are defined: parallel to $\nabla$POL ($k_{par}$) in the plane of the sheet, perpendicular to $\nabla$POL ($k_{per}$) in the plane of the sheet and normal to the plane of the sheet (growth in thickness, $k_{nor}$) [18]. However, this assumes prior information that defines the plane of the sheet. For volumetric growth, sheets may be an outcome of growth rather than a precondition. A more general formulation is, therefore,

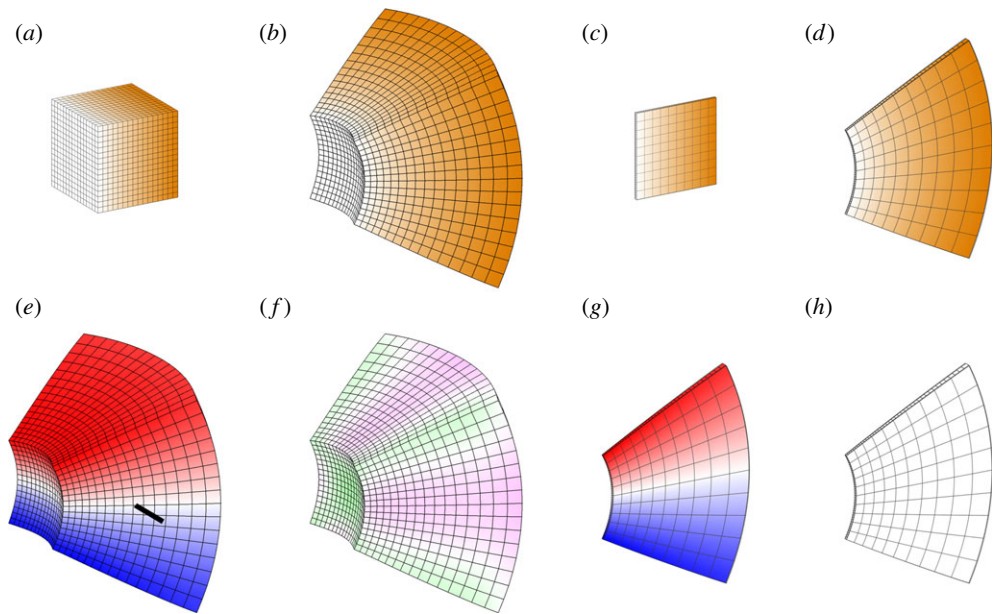

**Figure 9.** Linear gradient of isotropic growth. Specified linear growth rate varies from zero at the left to a maximum at the right. (*a,b*) Result of linear growth gradient applied to a cube, showing initial (*a*) and final (*b*) tissues. Orange colour scale indicates the specified volumetric growth rate. (*c,d*) Result of linear growth gradient applied to a thin square, showing initial (*c*) and final (*d*) tissues. (*e*) Vorticity about the indicated axis for the grown cube. (*f*) Residual strain for the grown cube. (*g*) Vorticity in the plane for the grown thin square. (*h*) Residual strain for the grown thin square (negligible). Angles are preserved (conformal mapping) and there is negligible residual strain and no buckling. Growth perpendicular to the plane is zero and the same random initial perturbation is applied as for the disc examples in figure 7. Orange represents the specified growth rate; magenta/green is the rate of residual compression/tension; red/blue is anticlockwise/clockwise rotation about the indicated axis.

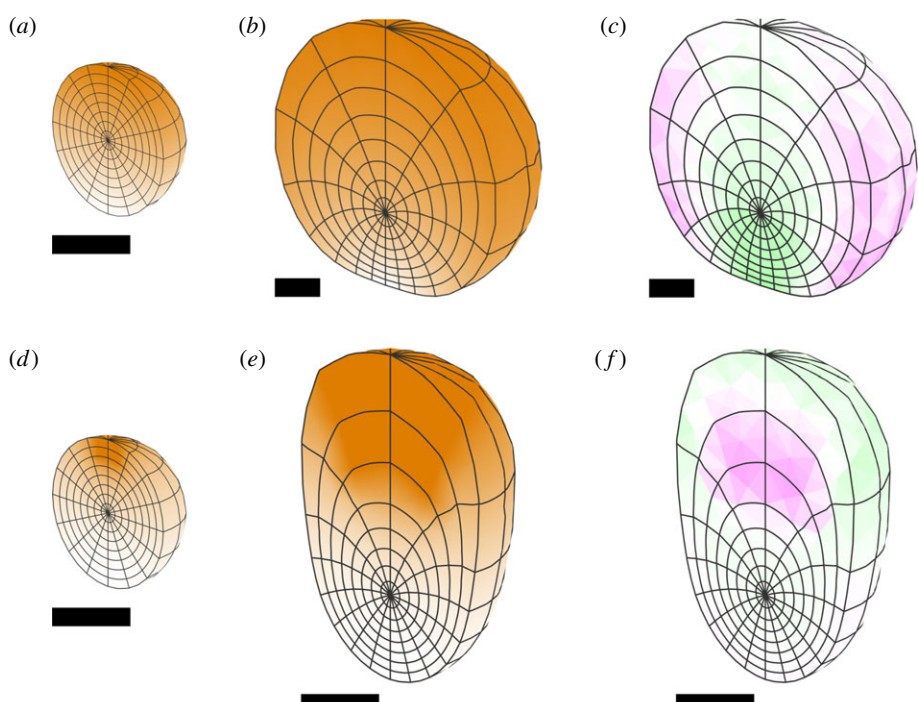

**Figure 10.** Gradients of isotropic specified growth in a sphere. (*a–c*) A linear gradient from bottom to top, showing a cutaway of the initial and final states, and residual strain in the final state. (*d–f*) Gradient set up from a source at the 'north pole' with diffusion and decay. All of the gradients are first set up and then remain fixed in the tissue without further diffusion. The resulting forms are similar if diffusion continues throughout. Orange represents specified growth rate, and magenta/green is the rate of residual compression/tension.

needed to specify growth orientations. With a gradient of a single morphogen POL, we may define a specified growth rate parallel to ∇POL ($k_{par}$), and a plane of directions perpendicular to this gradient. Within that plane there can be an isotropic growth rate $k_{per}$ (figure 11).

### 2.5.2. Growth patterns

We consider first the simplest form of a polarity field, a uniform linear gradient across a block, with a uniform growth rate (figure 12). The source of POL (plus-organizer or

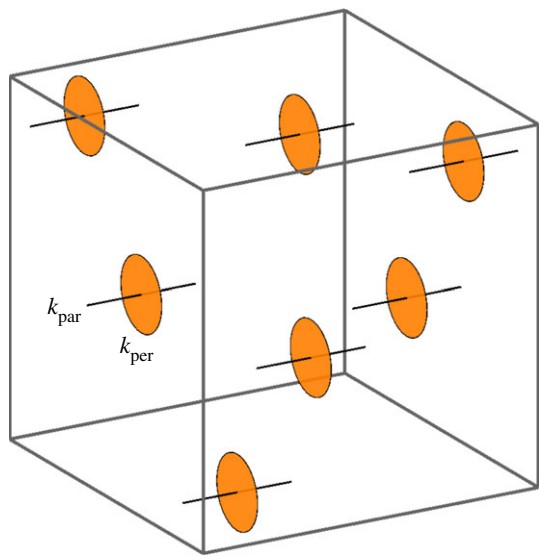

**Figure 11.** A single axis of polarization allows one growth rate to be defined parallel to it ($k_{par}$, black) and one to be defined that is isotropic in the perpendicular plane ($k_{per}$, orange).

+ORG, green) and sink (minus-organizer or −ORG, cyan) occupy opposite faces, giving a parallel polarity field (a). Uniform $k_{par}$ elongates the block along the field (b). Uniform $k_{per}$ generates a sheet (c). In both cases, residual strain is zero as there is neither directional nor regional conflict.

It is straightforward to generate a parallel polarity field in a rectilinear form such as a cube. However, with many forms, the polarity field will curve to follow the shape. For example, if organizers are positioned at opposite poles of a sphere, the polarity field will diverge or converge at the poles (figure 13a). If uniform specified growth is parallel to the polarity, the sphere elongates to form an ellipsoid with pointed ends. Conversely, if uniform specified growth is perpendicular to the polarity, the sphere deforms into a squashed disc concave at the poles. If a source of polarizer is used without a sink (i.e. no minus-organizer, figure 13b) but with the polarizer undergoing decay, uniform $k_{par}$ still gives a pointed ellipsoid, whereas uniform $k_{per}$ yields a cup-shaped disc.

A slightly different outcome is generated by using an equatorial (midplane) plus-organizer with polarizer that decays (figure 13c). Uniform $k_{par}$ still generates a pointed ellipsoid, but uniform $k_{per}$ generates a flat sheet. Thus a midplane organizer provides a simple mechanism for generating a sheet even from an initially curved volume. Using two orthogonal midplane organizers (figure 13d) from which POL diffuses, with positive $k_{per}$, this generates two intersecting sheets. Thus through directional conflict alone, it is possible to generate single sheets, or sheets with further sheets growing out of them.

## 2.6. Anisotropic specified growth along two orientation fields

In general, anisotropic growth can be defined locally by a set of three orthogonal axes (the *principal axes*) and three growth rates (the *principal growth rates*) along the respective axes. The gradient of a single polarity field defines only one axis, leaving all directions perpendicular to the gradient in a symmetrical relationship with each other. To break that

symmetry, a second polarity field is required, defining a second axis. The third axis can then be defined as their common perpendicular. We, therefore, introduce two polarizing morphogens, which we shall call POL and POL2, whose gradients ($\nabla$POL and $\nabla$POL2) specify two polarity fields. In general, these gradients will not be perpendicular to each other, but principal strain axes necessarily are. One approach to obtain perpendicular axes is to stipulate that $\nabla$POL is one of the principal axes, and the projection of $\nabla$POL2 onto the plane perpendicular to $\nabla$POL is the second. In this case, one of the polarities is primary ($\nabla$POL) and the other secondary ($\nabla$POL2). The specified growth rates along the three principal axes of the specified growth tensor are designated $k_{par}$, $k_{par2}$ and $k_{per}$ (figure 14).

### 2.6.1. Growth patterns

Consider a cube with two orthogonal polarity fields. The organizers for POL are on opposite faces (+ORG green, −ORG cyan), generating a parallel polarity field (figure 15a, black arrows). A gradient of POL2 is established through organizers on another pair of opposite faces (+ORG2 magenta, −ORG2, yellow), giving an orthogonal polarity field (figure 15b, blue arrows). The common perpendicular to both polarity fields is indicated with orange lines (figure 15c). If specified growth rates are uniform, with $k_{par} > k_{par2} > k_{per}$ we get the block shown in figure 15d, in which length > height > width. Blocks of different dimensions may be generated by varying the three values of specified growth and the duration. Because growth is uniform and the polarity fields are parallel, there is no growth conflict and residual strains are zero.

With an initial sphere, the shape changes are less intuitively predictable because of directional conflicts resulting from curvature of the polarity field. Also, the polarity fields for POL and POL2 may not be everywhere orthogonal. Consider a case where a horizontal midplane +ORG defines a polarity field for POL (figure 16a,b); whereas the organizers for POL2 are at opposite points on the equator (figure 16c,d). If $k_{par} = 0$, and there is uniform $k_{par2} = k_{per}$, we obtain a disc-shaped sheet (figure 16e), as specified growth is isotropic in the plane perpendicular to POL.

If $k_{par} = k_{per} = 0$, and $k_{par2}$ is uniform, we obtain a solid cylinder with POL2 arrows running from one end to the other (figure 16f). However, if $k_{par} = 0$ and $k_{par2} > k_{per}$, we obtain an oval sheet (figure 16g). From a genetic perspective, if we consider the latter as a wild-type shape, then setting $k_{per} = 0$ is equivalent to radialized mutant which has lost the ability to generate a sheet. Conversely, setting $k_{par2} = k_{per}$ is equivalent to a symmetrical mutant in which the length and breadth of the sheet are identical.

Leaf-like shapes can be generated by restricting $k_{per}$ in the region destined to be the base of the leaf. Two different ways of doing this are shown in figure 16h–k, yielding two different leaf shapes. The smooth gradient illustrated in (h) yields the slender taper shown in (i), while setting $k_{par2}$ to zero near one end and 1 in the remainder gives a more defined petiole shape (j–k).

To illustrate how this approach can be used to model processes that could not be captured using sheets alone, we modelled the formation of solid outgrowths from a leaf, observed in the *kanadi1 kanadi2* double mutant of *Arabidopsis* [37]. These outgrowths arise in leaf primordia at positions in the epidermis where planar polarity, as revealed by PIN1,

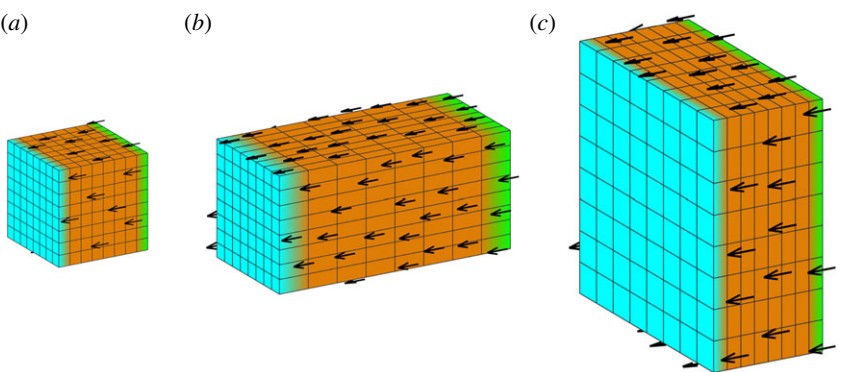

**Figure 12.** Anisotropic growth of a solid cube with a single polarizer. (*a*) Initial cube with polarity field. (*b*) Result of uniform positive $k_{par}$ and zero $k_{per}$: an elongated block. (*c*) Result of uniform positive $k_{per}$ and zero $k_{par}$: a sheet. $+$ORG is green, $-$ORG is cyan, and orange indicates uniform $k_{par}$ or $k_{per}$.

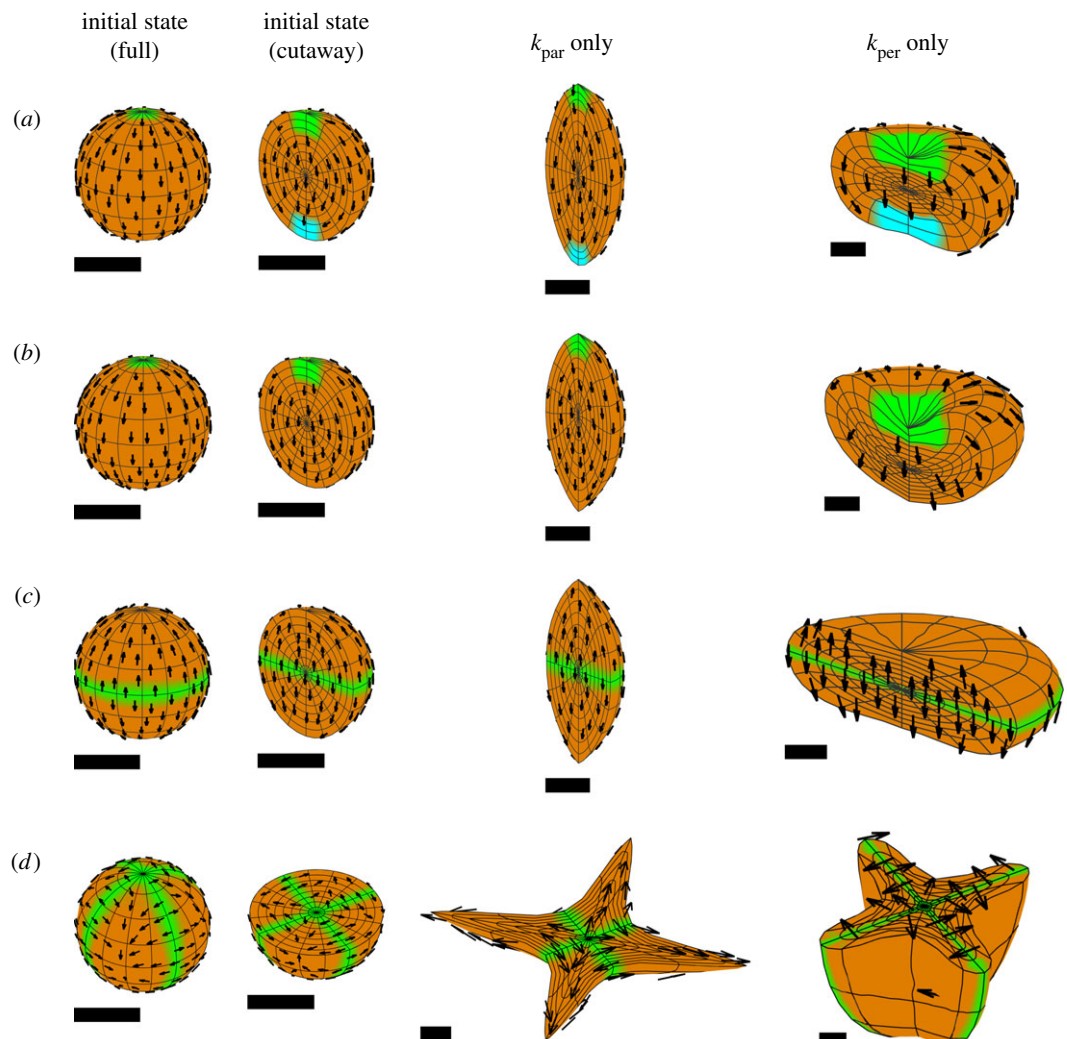

**Figure 13.** Anisotropic growth of a sphere. Growth is uniform, and either parallel or perpendicular to POL. (*a*) POL is created by diffusion between a source ($+$ORG, green) and a sink ($-$ORG, cyan) in small regions at opposite poles. Positive $k_{par}$ creates a pointed ellipsoid. Positive $k_{per}$ creates an indented disc. (*b*) POL is created by diffusion and decay from a source at the upper pole. Positive $k_{par}$ gives a pointed ellipsoid. Positive $k_{per}$ gives a cup. (*c*) POL is produced by a source in an equatorial region, diffuses and decays. Positive $k_{par}$ gives a pointed ellipsoid. Positive $k_{per}$ gives a flat sheet. (*d*) POL is produced by two vertical midplanes at right angles to each other. Positive $k_{par}$ gives a star shape, whose points are in the diagonal directions. Positive $k_{per}$ gives two interpenetrating sheets. The cutaway views here show the lower half of the tissue. The upper half is symmetric. Orange represents the specified growth rate, green $+$ORG and cyan $-$ORG. The arrow fields can be more clearly seen under enlargement.

converges. A subepidermal strand of PIN polarity also forms and runs through the centre of the outgrowth. To model this process, we modified the model used in figure 16*h*,*i*, by growing it to a primordial stage, and then introducing a $-$ORG for POL2 to create a site of planar polarity convergence (figure 17*a*). We also created a strand of $+$ORG for POL, running from the convergence site towards the midplane (figure 17*b*). Both $k_{par2}$ and $k_{per}$ were also increased in the

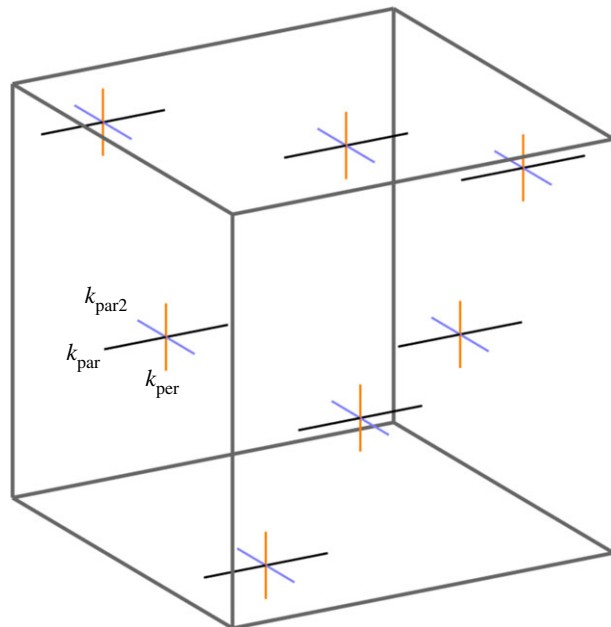

**Figure 14.** Two axes of polarization allow one growth rate to be defined parallel to each ($k_{par}$, black, and $k_{par2}$, blue) and one to be defined that is perpendicular to both ($k_{per}$, orange).

neighbourhood of the +ORG strand (figure 17*c,d*). With these assumptions an outgrowth formed, centred on the +ORG strand (figure 17*e,f,h*). This model is beyond the capability of the earlier version of GFtbox, which was limited to sheet-like meshes which could vary in thickness, but did not support changes of form beyond that.

## 3. Discussion

The volumetric growth modelling framework presented here is more general than that previously developed to model growth of tissue sheets [18]. Rather than presupposing a sheet-like configuration, with growth being defined parallel to the tissue plane or normal to it, sheets and other tissue forms are generated as an outcome of growth. A major difference between sheet and bulk tissues is that the growth of bulk tissue is more geometrically constrained. Every point of a sheet has empty space close by, into which tissue can move to reduce residual strain generated by growth conflicts. By contrast, with bulk tissues regions may be constrained by material all around them, reducing the possibilities for conflict resolution by geometrical deformation. Thus, a radially increasing rate of isotropic growth for a circular sheet can lead to a reduction in residual strain by buckling to form a wavy edge, whereas the same pattern of growth for a solid sphere leads to higher levels of residual strain and no shape change. Similarly, a linear gradient of increasing specified isotropic growth across a square sheet generates a conformal deformation with little or no accumulation of residual strain, whereas a similar pattern of growth for a cube is non-conformal and generates some residual strain.

For many plant tissues, such as the shoot apex or hypocotyl, surgical incisions have shown that the epidermal layer is under residual tension whereas the internal tissue is under residual compression [36]. If specified growth is isotropic, this situation would correspond to specified growth rates being higher for internal than for outer tissue layers.

However, epidermal tension can also be generated by uniform anisotropic growth oriented radially, showing that levels of residual tension alone cannot be used to infer patterns of specified growth. Orientations as well as the level of residual stresses could be helpful in distinguishing between different possibilities. Incisions could be implemented in GFtbox, and have been for sheet-like models, but we have not yet implemented this for volumetric models.

The shape deformations generated through differential specified isotropic growth depend on both the initial starting shape and the pattern of growth. For example, whereas a radially increasing specified growth rate leads to no shape change for a sphere, it leads to a cube forming outgrowths at the corners. This is because the corners are less constrained by surrounding bulk tissue. Conversely, if the specified growth rate decreases radially, the corners of a cube become less sharp, resolving some of the residual strain. The pattern of growth is also important. A linear gradient of specified growth rate across a sphere (from pole to pole) leads to the formation of a dimpled shape, whereas a more restricted region of higher specified growth leads to an egg shape. In both cases, the region of high growth comes to dominate the tissue more and more. This raises the question of how meristematic domains, such as those in the shoot or root apex, maintain a relatively fixed size in the face of growth.

Although differential isotropic specified growth can create a range of shape deformations, some are much harder to achieve. For example, transforming a sphere into a sheet is not straightforward through isotropic specified growth alone (though it may be possible). By contrast, the transformation can be readily achieved through specified anisotropic growth oriented by a polarity field. For example, if specified growth is high perpendicular to a polarity field defined by a midplane organizer, a sphere deforms into a sheet. Growth parallel to the polarity corresponds to growth in sheet thickness whereas growth perpendicular to the polarity corresponds to growth in the plane of the sheet. This view is consistent with experimental studies on genes controlling leaf abaxial/adaxial asymmetry [38]. These genes establish a midplane domain of *WOX* expression which is thought to be critical for establishing planar growth [39,40]. From a cellular perspective, the higher specified growth rate perpendicular to the polarity field corresponds to cell walls being less reinforced in these directions than parallel to the field. However, whether a polarity field of this type exists and how it may influence cell wall stiffness remains to be established [41].

Although a single polarity field allows for many shape transformations, it only allows two of the three components of a specified strain tensor to be defined. Thus, it would not be straighforward to generate a leaf that is much longer than it is wide. Such transformations can be readily achieved through inclusion of a second polarity field, orthogonal to the first. Applying such fields to a sphere allows an elongated leaf-like sheet to be generated, with one polarity field becoming oriented in the plane of the sheet (planar polarity) and the other oriented normal to the sheet (orthoplanar polarity). Polarly localized proteins such as PIN and BASL provide evidence for a planar polarity field in leaves [42,43]. Loss of the orthoplanar polarity field (e.g. midplane organizer) leads to reduced outgrowth of the blade or lamina [39,40]. Ectopic

royalsocietypublishing.org/journal/rsob    Open Biol. **9**: 190057

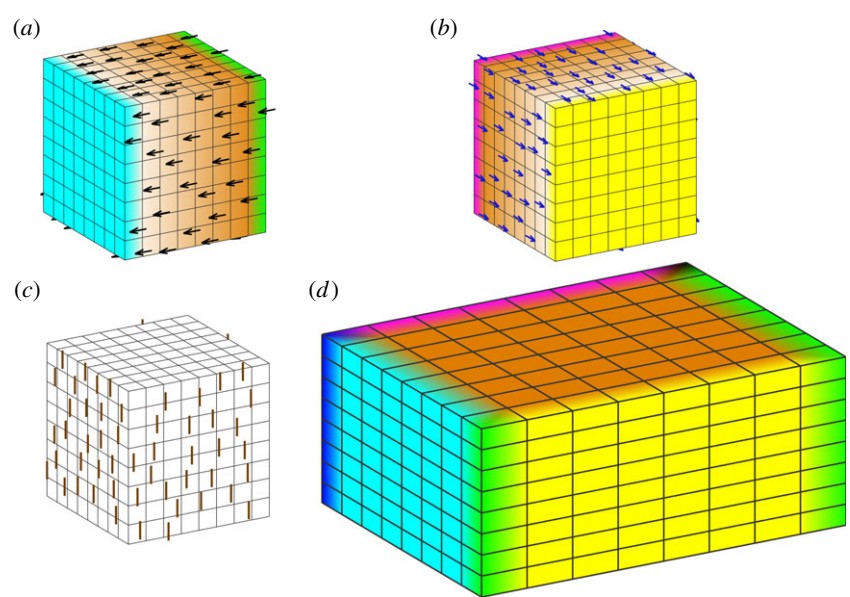

**Figure 15.** Doubly anisotropic growth of an initial cube. (*a*) $+$ORG (green) and $-$ORG (cyan) with $\nabla$POL (black arrows). (*b*) $+$ORG2 (magenta) and $-$ORG2 (yellow) with $\nabla$POL2 (blue arrows). (*c*) The common perpendicular of the two gradient fields (orange lines). (*d*) The result of uniform specified growth of different amounts along each of the three direction fields, $k_{par} > k_{par2} > k_{per}$.

**Figure 16.** Generation of shapes from a sphere with two polarizers and uniform growth rates. (*a,b*) A midplane $+$ORG (green) defines a polarity field for POL. (*c,d*) Organizers for POL2 are at the front left and rear right of the equatorial plane ($+$ORG2, magenta, $-$ORG2, yellow). (*e*) If $k_{par} = 0$, and there is uniform $k_{par2} = k_{per}$, we obtain a disc-shaped sheet. (*f*) If $k_{par} = k_{per} = 0$, and $k_{par2}$ is uniform, we obtain an elongated cylinder. (*g*) If $k_{par} = 0$ and $k_{par2}$ is greater than $k_{per}$, we obtain an oval sheet. (*h*) Giving $k_{per}$ a gradient from left to right in the model for (*g*) yields a leaf-like shape (*i*). (*j*) Setting $k_{per}$ instead to zero in a neighbourhood of $+$ORG2 gives a different leaf-like shape (*k*). Orange indicates the growth rate, and specifically $k_{per}$ in (*h–k*). All except (*a,c*) are shown sectioned in the equatorial plane. POL2 arrows are shown in (*e,f,g,i,k*).

**Figure 17.** The model of figure 16*h,i*, modified by initiating an outgrowth. (*a – d*) At the time when the outgrowth is initiated, in cross-section. (*a*) $\nabla$POL2 and its plus and minus organizers, (*b*) $\nabla$POL and its plus organizer, (*c*) $k_{par2}$, (*d*) $k_{per}$. (*e,f,g*) The subsequent development, showing $k_{per}$ and $\nabla$POL2.

activation of a strand of the midplane organizer can lead to the formation of radial outgrowths, as we show by modelling the formation of leaf outgrowths in the *kanadi1 kanadi2* double mutant of *Arabidopsis*. There is evidence for a radial polarity field in roots [44,45], but evidence for a midplane polarity field has yet to be obtained.

Dynamic subdivision is more complex in the volumetric finite-element framework. In two dimensions, there is a small repertoire of local transformations that eliminate low-quality finite elements. However, in three dimensions, the number of cases to consider is much larger, and the transformations are much more complex. Moreover, validity constraints (e.g. always generating the same geometric class of element) complicate the task of locally transforming the mesh to improve quality. Our implementation is in this respect still incomplete: we have implemented only the simpler cases of dynamic subdivision and not incorporated elision of short edges. This need not imply that division of biological cells is more difficult to achieve in volumes than sheets. This is because biological cell division is not constrained by having to always generate the same geometric class. To model biological cells with a finite-element framework, multiple elements would be needed for each cell, introducing additional complexity.

In conclusion, our analysis shows that the finite-element framework used to model tissue sheets can be extended to model volumetric growth. This extension highlights the additional constraints brought about through growth in three dimensions, and also provides a basis for studying the generation of tissue sheets or other conformations that have previously been taken as starting points.

Data accessibility. This article has no additional data.

Competing interests. We declare we have no competing interests.

Funding. This work was supported by grants from the Biotechnology and Biological Sciences Research Council (grant nos. BB/M023117/1, BB/L008920/1, BB/P020747/1 and BBS/E/J/000PR9787) and ERC (grant no. Carnomorph, 323028) awarded to E.C.

Acknowledgements. We thank John Fozard, Karen Lee, Jie Cheng and the anonymous referees for critical reading of the manuscript.

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
