## [Reviewer comments · Open Biology]

Review History

RSOB-19-0057.R0 (Original submission)

Review form: Reviewer 1

Recommendation

Accept with minor revision (please list in comments)

Are each of the following suitable for general readers?

- a) **Title**
Yes
- b) **Summary**
Yes
- c) **Introduction**
Yes

Is the length of the paper justified?

Yes

Should the paper be seen by a specialist statistical reviewer?

No

Is it clear how to make all supporting data available?

Yes

Is the supplementary material necessary; and if so is it adequate and clear?

Yes

Do you have any ethical concerns with this paper?

No

Comments to the Author

The authors present a framework for modeling 3-dimensional growth with finite elements, an extension of their previous work on modeling continuous sheets of tissue. As in their previous work, they combine two classes of morphogens, polarizers that determine tissue polarity and directions in the tissue, and growth factors that determine growth amounts in relation to these directions.

Although one can see the potential of the framework to develop a wide variety of biological shapes, it still would be nice to have included at least one concrete 3D example, other than the leaf-like shapes in Fig 16 that could have been made with the previous framework.

I like the way they have addressed subdivision, and the discussion of the importance of removing bad tetra which would be harmful to convergence of the simulations, even if not implemented. The way they extend polarizers from their 2D sheets to 3D is very natural. I also like comparison of some of the test cases with analytical solutions.

The authors should spend some time discussion how mechanically realistic their model is. In a real cellular tissue, growth is driven by turgor pressure that puts stress on the cell wall, which is then relaxed by cell wall remodelers. Thus growth is stress relaxation process. If cells in the middle of the tissue have more specified growth, wouldn't they compress their neighbors? Is this realistic? Wouldn't it be better to apply pressure to boundary of the structure (assuming uniform turgor pressure) and release the stress non-uniformly (or anisotropically)? Of course this requires the preservation of residual stresses in the model.

The authors should discuss what happens to residual stress in the simulations, and if/how it is propagated during subdivision. I would also be interested to know how this is done for the butterfly subdivision in the 2D case.

Specific comments:

Abstract: The authors write "comprehensive framework for modelling volumetric growth in three dimensions has been lacking".

Both Bassel et al. 2014 and Boudon et al 2015 present models of growing 3D cellular tissues, albeit without cell division.

Page 2: It might be good to explicitly say that first order elements are linear.

Page 6: In the discussion of the difficulty of dynamically subdividing hexahedra, the authors

assume that the mesh has to be all of the same type of elements. In engineering work, a mesh may be mostly hexahedra, but with some prisms inserted to maintain mesh quality.

Page 6: On subdivision, the authors comment that butterfly subdivision is better than flat, but does this not complicate the propagation of residual stresses? Or are residual stresses just released at each time point?

Page 9: What does it mean for internal parts of a 3D structure to be “specified” to grow more? Is not growth a stress relaxation process? If an organ has uniform turgor, how can areas inside grow “more” and compress their neighbors?

Page 9: The authors write that 8b induces a shape change but 6b does not. Isn't this because you are imposing a non-uniform spherical growth field on a cube? Wouldn't the same happen with a cube shaped growth field on a sphere?

Page 15: Transverse isotropic polarity field?

Page 19: The authors write “this situation would correspond to specified growth rates being higher for internal compared to outer tissue layers”

If growth is a stress release process, the inside can release all of its residual stress, and still may not grow if it is constrained by the outer layer. In this sense growth itself does not put stress on the outer layer, turgor does, and it is more a matter of how much of the stress from turgor is borne by inner layers.

Page 19: It would be interesting to see if the two scenarios (isotropic vs radial anisotropic growth) produce the same reaction to cutting experiments. Can't the cutting experiments be done in the model?

Some typos:

Intro: “mathematically modelling” should be “mathematically model”.

Figure 7: a,,b,c not referenced in caption.

Review form: Reviewer 2

Comments to the Author

This manuscript presents a framework for computational modelling of volumetric elastic tissue growth, using a finite element approach. This framework allows for anisotropic and/or non-uniform specified growth, included growth in response to one or more polarity fields present in a tissue. The authors use several informative and well-chosen examples to illustrate how the growth of 'bulk' tissues can differ qualitatively from that of sheets. The authors have implemented their computational framework within a MATLAB package, which has been made available for download.

Overall I found this manuscript to be clearly written. It contains a number of computational findings that are of generic interest to those working on tissue morphogenesis from a theoretical and/or experimental perspective. I have general and specific comments and queries regarding the presentation and the work itself.

General comments

* Throughout the bulk of the present work, the authors rarely refer explicitly to biological applications. For example, when discussing anisotropic specified growth along one field of orientations, the authors refer to generic models where there are one or more polarity fields, but do not discuss concrete examples of such tissues. Including such references could help more closely tie the work to biology.

* On the other hand, the authors refer extensively to a previous publication (Kenaway et al, 2011) for technical details such as constitutive equations. It would be beneficial to have the constitutive equations at least briefly defined in the present manuscript - this would not take up much further space, and would save readers from having to look up another paper just to find out what equations are being solved in the present work.

* From a computational perspective, what is the performance of the code? For example, do each of the simulations presented in Figures 6-16 take seconds, or minutes, or hours to run? Including a brief indication of this in the supplementary material would be beneficial to readers.

Specific comments

p.3 "Dynamic Refinement" - The authors clearly describe dynamic remeshing of finite elements that are very large or thin. However, it wasn't quite clear to me how this process is implemented across an entire mesh - do the authors randomise the order in which finite elements are refined after each time step? Does this order matter for the resulting overall mesh?

p.13 "Initial and final canvas" - Here and elsewhere in the manuscript, "canvas" seems to have a technical meaning, but I'm not aware what this is (maybe it is domain-specific); can the authors clarify?

Typos

p.3 "moprhogen" -> "morphogen"

p.8 "with an initially circular sheet (Fig 7(a))." -> "with an initially circular sheet (Fig 7(a))."

p.25 "movies of some the developments" -> "movies of some of the developments"

p.28 "edges of tetrahedral mesh" -> "edges of the tetrahedral mesh"

p.29 "vertexes" -> "vertices"

Decision letter (RSOB-19-0057.R0)

08-Apr-2019

Dear Professor Coen

We are pleased to inform you that your manuscript RSOB-19-0057 entitled "Volumetric finite element modelling of biological growth" has been accepted by the Editor for publication in Open Biology. The reviewer(s) have recommended publication, but also suggest some minor revisions

to your manuscript. Therefore, we invite you to respond to the reviewer(s)' comments and revise your manuscript.

Please submit the revised version of your manuscript within 14 days. If you do not think you will be able to meet this date please let us know immediately and we can extend this deadline for you.

- 1) A text file of the manuscript (doc, txt, rtf or tex), including the references, tables (including captions) and figure captions. Please remove any tracked changes from the text before submission. PDF files are not an accepted format for the "Main Document".
- 2) A separate electronic file of each figure (tiff, EPS or print-quality PDF preferred). The format should be produced directly from original creation package, or original software format. Please note that PowerPoint files are not accepted.
- 3) Electronic supplementary material: this should be contained in a separate file from the main text and meet our ESM criteria (see <http://royalsocietypublishing.org/instructions-authors#question5>). All supplementary materials accompanying an accepted article will be treated as in their final form. They will be published alongside the paper on the journal website and posted on the online figshare repository. Files on figshare will be made available approximately one week before the accompanying article so that the supplementary material can be attributed a unique DOI.

Online supplementary material will also carry the title and description provided during submission, so please ensure these are accurate and informative. Note that the Royal Society will not edit or typeset supplementary material and it will be hosted as provided. Please ensure that the supplementary material includes the paper details (authors, title, journal name, article DOI). Your article DOI will be 10.1098/rsob.2016[last 4 digits of e.g. 10.1098/rsob.20160049].

- 4) A media summary: a short non-technical summary (up to 100 words) of the key findings/importance of your manuscript. Please try to write in simple English, avoid jargon, explain the importance of the topic, outline the main implications and describe why this topic is newsworthy.

Images

Data-Sharing

It is a condition of publication that data supporting your paper are made available. Data should be made available either in the electronic supplementary material or through an appropriate repository. Details of how to access data should be included in your paper. Please see <http://royalsocietypublishing.org/site/authors/policy.xhtml#question6> for more details.

Data accessibility section

Sincerely,

The Open Biology Team
<mailto:openbiology@royalsociety.org>

Reviewer(s)' Comments to Author:

Referee: 1

Comments to the Author(s)

The authors present a framework for modeling 3-dimensional growth with finite elements, an extension of their previous work on modeling continuous sheets of tissue. As in their previous work, they combine two classes of morphogens, polarizers that determine tissue polarity and directions in the tissue, and growth factors that determine growth amounts in relation to these directions.

Although one can see the potential of the framework to develop a wide variety of biological shapes, it still would be nice to have included at least one concrete 3D example, other than the leaf-like shapes in Fig 16 that could have been made with the previous framework.

I like the way they have addressed subdivision, and the discussion of the importance of removing bad tetra which would be harmful to convergence of the simulations, even if not implemented. The way they extend polarizers from their 2D sheets to 3D is very natural. I also like comparison of some of the test cases with analytical solutions.

The authors should spend some time discussion how mechanically realistic their model is. In a real cellular tissue, growth is driven by turgor pressure that puts stress on the cell wall, which is then relaxed by cell wall remodelors. Thus growth is stress relaxation process. If cells in the

middle of the tissue have more specified growth, wouldn't they compress their neighbors? Is this realistic? Wouldn't it be better to apply pressure to boundary of the structure (assuming uniform turgor pressure) and release the stress non-uniformly (or anisotropically)? Of course this requires the preservation of residual stresses in the model.

The authors should discuss what happens to residual stress in the simulations, and if/how it is propagated during subdivision. I would also be interested to know how this is done for the butterfly subdivision in the 2D case.

Specific comments:

Abstract: The authors write "comprehensive framework for modelling volumetric growth in three dimensions has been lacking".

Both Bassel et al. 2014 and Boudon et al 2015 present models of growing 3D cellular tissues, albeit without cell division.

Page 2: It might be good to explicitly say that first order elements are linear.

Page 6: In the discussion of the difficulty of dynamically subdividing hexahedra, the authors assume that the mesh has to be all of the same type of elements. In engineering work, a mesh may be mostly hexahedra, but with some prisms inserted to maintain mesh quality.

Page 6: On subdivision, the authors comment that butterfly subdivision is better than flat, but does this not complicate the propagation of residual stresses? Or are residual stresses just released at each time point?

Page 9: What does it mean for internal parts of a 3D structure to be "specified" to grow more? Is not growth a stress relaxation process? If an organ has uniform turgor, how can areas inside grow "more" and compress their neighbors?

Page 9: The authors write that 8b induces a shape change but 6b does not. Isn't this because you are imposing a non-uniform spherical growth field on a cube? Wouldn't the same happen with a cube shaped growth field on a sphere?

Page 15: Transverse isotropic polarity field?

Page 19: The authors write "this situation would correspond to specified growth rates being higher for internal compared to outer tissue layers"

If growth is a stress release process, the inside can release all of its residual stress, and still may not grow if it is constrained by the outer layer. In this sense growth itself does not put stress on the outer layer, turgor does, and it is more a matter of how much of the stress from turgor is borne by inner layers.

Page 19: It would be interesting to see if the two scenarios (isotropic vs radial anisotropic growth) produce the same reaction to cutting experiments. Can't the cutting experiments be done in the model?

Some typos:

Intro: "mathematically modelling" should be "mathematically model".

Figure 7: a,,b,c not referenced in caption.

Referee: 2

Comments to the Author(s)

This manuscript presents a framework for computational modelling of volumetric elastic tissue growth, using a finite element approach. This framework allows for anisotropic and/or non-uniform specified growth, included growth in response to one or more polarity fields present in a tissue. The authors use several informative and well-chosen examples to illustrate how the growth of 'bulk' tissues can differ qualitatively from that of sheets. The authors have implemented their computational framework within a MATLAB package, which has been made available for download.

Overall I found this manuscript to be clearly written. It contains a number of computational findings that are of generic interest to those working on tissue morphogenesis from a theoretical and/or experimental perspective. I have general and specific comments and queries regarding the presentation and the work itself.

General comments

* Throughout the bulk of the present work, the authors rarely refer explicitly to biological applications. For example, when discussing anisotropic specified growth along one field of orientations, the authors refer to generic models where there are one or more polarity fields, but do not discuss concrete examples of such tissues. Including such references could help more closely tie the work to biology.

* On the other hand, the authors refer extensively to a previous publication (Kenaway et al, 2011) for technical details such as constitutive equations. It would be beneficial to have the constitutive equations at least briefly defined in the present manuscript - this would not take up much further space, and would save readers from having to look up another paper just to find out what equations are being solved in the present work.

* From a computational perspective, what is the performance of the code? For example, do each of the simulations presented in Figures 6-16 take seconds, or minutes, or hours to run? Including a brief indication of this in the supplementary material would be beneficial to readers.

Specific comments

p.3 "Dynamic Refinement" - The authors clearly describe dynamic remeshing of finite elements that are very large or thin. However, it wasn't quite clear to me how this process is implemented across an entire mesh - do the authors randomise the order in which finite elements are refined after each time step? Does this order matter for the resulting overall mesh?

p.13 "Initial and final canvas" - Here and elsewhere in the manuscript, "canvas" seems to have a technical meaning, but I'm not aware what this is (maybe it is domain-specific); can the authors clarify?

Typos

p.3 "moprhogen" -> "morphogen"

p.8 "with an initially circular sheet (Fig 7(a))." -> "with an initially circular sheet (Fig 7(a))."
p.25 "movies of some the developments" -> "movies of some of the developments"
p.28 "edges of tetrahedral mesh" -> "edges of the tetrahedral mesh"
p.29 "vertexes" -> "vertices"

Author's Response to Decision Letter for (RSOB-19-0057.R0)

See Appendix A.

Decision letter (RSOB-19-0057.R1)

03-May-2019

Dear Professor Coen

We are pleased to inform you that your manuscript entitled "Volumetric finite element modelling of biological growth" has been accepted by the Editor for publication in Open Biology.

Sincerely,

The Open Biology Team
mailto:openbiology@royalsociety.org

Appendix A

Volumetric finite element modelling of biological growth

Response to reviewers

Richard Kennaway and Enrico Coen
Cell and Developmental Biology, John Innes Centre, Norwich, U.K.

10 Apr 2019

We would like to thank both referees for their very helpful and constructive comments. We address the points raised below.

Page numbers in our responses refer to the final resubmitted version, but these may change again in the publication process.

Referee 1

1. *Although one can see the potential of the framework to develop a wide variety of biological shapes, it still would be nice to have included at least one concrete 3D example, other than the leaf-like shapes in Fig 16 that could have been made with the previous framework.*

We have added another model to illustrate a development that is beyond the capabilities of the previous version of GFtbox. This is illustrated in Figure 17, and described in the text on pp.18–20, in the last paragraph before the Discussion section.

To illustrate how this approach can be used to model processes that could not be captured using sheets alone, we modelled the formation of solid outgrowths from a leaf, observed in the *kanadi1 kanadi2* double mutant of *Arabidopsis* [36]. These outgrowths arise in leaf primordia at positions in the epidermis where planar polarity, as revealed by PIN1, converges. A subepidermal strand of PIN polarity also forms and runs through the centre of the outgrowth. To model this process, we modified the model used in Figure 16(h,i), by growing it to a primordial stage, and then introducing a –ORG for POL2 to create a site of planar polarity convergence (Figure 17(a)). We also created a strand of +ORG for POL, running from the convergence site towards the midplane (Figure 17(b)). Both k_{par2} and k_{per} were also increased in the neighbourhood of the +ORG strand (Figure 17(c,d)). With these assumptions an outgrowth formed, centred on the +ORG strand (Fig 17(e,f,g)). This model is beyond the capability of the earlier version of GFtbox, which was limited to sheet-like meshes which could vary in thickness, but did not support changes of form beyond that.

It is also briefly mentioned in the Discussion on p.21 and alluded to in the phrase “and outgrowths” at the foot of p.2.

2. *The authors should spend some time discussion how mechanically realistic their model is. In a real cellular tissue, growth is driven by turgor pressure that puts stress on the cell wall, which is then relaxed by cell wall remodelors. Thus growth is stress relaxation process. If cells in the*

middle of the tissue have more specified growth, wouldn't they compress their neighbors? Is this realistic? Wouldn't it be better to apply pressure to boundary of the structure (assuming uniform turgor pressure) and release the stress non-uniformly (or anisotropically)? Of course this requires the preservation of residual stresses in the model.

The authors should discuss what happens to residual stress in the simulations, and if/how it is propagated during subdivision. I would also be interested to know how this is done for the butterfly subdivision in the 2D case.

The referee raises a valuable question as to the relationship between our framework and cellular behaviour. We have added text to the Introduction on page 2 relating the concept of specified growth to the underlying mechanisms of turgor and the stretching and infilling of cell walls:

The physical mechanism of growth of plant tissue is generally held to be that cells enlarge by turgor pressure acting to stretch cell walls whose elastic stiffness has been reduced, which are then reinforced by the addition of wall material. In our framework, the specified growth of each cell-sized region during a time step corresponds to the amount by which the cell would incrementally yield in this way, if it were unconstrained by the surrounding tissue. The resultant growth of each region is then its equilibrium conformation given the constraint that the whole tissue remains continuous.

The discarding of residual strain after each time step amounts to treating the resultant deformation as an irreversible plastic flow. Thus the framework treats growth as a viscoelastic process [1]. The amount of plasticity may be reduced by retaining all or part of the residual strain.

The following paragraph is added to p.9:

In cellular terms, the notion of residual compression does not mean that plant cell walls themselves are under compression, only that the region of tissue is held at a smaller size than it would otherwise take without the external constraint. Thus cell walls will still be under tension due to osmotic pressure, even in the presence of residual compression.

For the butterfly subdivision issue, see our response to his mention of subdivision below.

Specific comments:

3. *Abstract: The authors write “comprehensive framework for modelling volumetric growth in three dimensions has been lacking”. Both Bassel et al. 2014 and Boudon et al 2015 present models of growing 3D cellular tissues, albeit without cell division.*

We consider our framework to be more general than either of these references. They both use membrane (two-dimensional) elements to model tissue as an assemblage of cell walls, and as the referee notes, they do not model cell division. This limits the amount of growth that they are able to model. By modelling the tissue at a higher level as a continuous substance, we are able to model growth over large changes in shape and volume, in which the number of cells would be too great to model individually. We have modified the quoted phrase in the abstract (p.1) to be more exact:

Differential growth is the driver of tissue morphogenesis in plants, and also plays a fundamental role in animal development. Although the contributions of growth to shape change have been captured through modelling tissue sheets or isotropic volumes, **a framework for modelling both isotropic and anisotropic volumetric growth in three dimensions over large changes in size and shape** has been lacking.

4. *Page 2: It might be good to explicitly say that first order elements are linear.*

We have added the word “linear” to the mention of first-order elements on p.3:

For simplicity of coding, we have implemented only first-order **(linear)** elements

5. *Page 6: In the discussion of the difficulty of dynamically subdividing hexahedra, the authors assume that the mesh has to be all of the same type of elements. In engineering work, a mesh may be mostly hexahedra, but with some prisms inserted to maintain mesh quality.*

We have noted the possibility of meshes of mixed types of element on page 6:

For meshes made of pentahedra (warped triangular prisms), and hexahedra (warped cubes), dynamic remeshing is more difficult. It is easy to subdivide every element simultaneously into smaller elements, but to refine some arbitrary subset, while maintaining consistency across shared faces, is not necessarily possible, **unless one allows meshes that mix different shapes of element.**

6. *Page 6: On subdivision, the authors comment that butterfly subdivision is better than flat, but does this not complicate the propagation of residual stresses? Or are residual stresses just released at each time point?*

We do not modify residual strains during butterfly subdivision. The purpose of butterfly subdivision is to better approximate a smooth surface. The displacement of vertices prescribed by butterfly subdivision is not a deformation of the elements (which would change their state of strain), but a remeshing of the smooth-surfaced shape, with the same strain field. We have added a paragraph to this effect at the end of the section on dynamic refinement (p.6):

The displacement of vertexes prescribed by the butterfly method should not be thought of as a deformation of the finite elements, but as a remeshing that better approximates the smooth-surfaced volume that is being modelled. Therefore, if residual strain is present in the elements, it retains its values during subdivision. The volume of a divided element at the surface will increase slightly where the surface is convex and decrease where it is concave, thus changing its total strain energy, but the amount is slight, and comparable to the finite accuracy of the finite element method itself.

7. *Page 9: What does it mean for internal parts of a 3D structure to be “specified” to grow more? Is not growth a stress relaxation process? If an organ has uniform turgor, how can areas inside grow “more” and compress their neighbors?*

See our response to point 2 above.

8. *Page 9: The authors write that 8b induces a shape change but 6b does not. Isn't this because you are imposing a non-uniform spherical growth field on a cube? Wouldn't the same happen with a cube shaped growth field on a sphere?*

Yes, the sphere retains its overall shape because its geometric symmetry is the same as the growth field's symmetry. We have clarified this point on page 9:

The sphere shows no overall change of shape in these examples (it remains a sphere), due to the common spherical symmetry of the tissue and the growth pattern.

A cube-shaped growth field (that is, isotropic growth rate proportional to maximum initial absolute coordinate value) does indeed produce growth with cubical rather than spherical symmetry. On the other hand, a cube-shaped growth field applied to the cube (not included in the paper) still causes its corners to bulge outwards, as they have more space around them to relieve the strain.

9. *Page 15: Transverse isotropic polarity field?*

We have clarified the wording on that page (now page 12 due to movement of figures):

To model anisotropic specified growth of sheets, three orthogonal axialities are defined: parallel to ∇POL (k_{par}) in the plane of the sheet, perpendicular to ∇POL (k_{per}) in the plane of the sheet, and normal to the plane of the sheet (growth in thickness, k_{nor}) [2]. However, this assumes prior information that defines the plane of the sheet. For volumetric growth, sheets may be an outcome of growth rather than a precondition. A more general formulation is therefore needed to specify growth orientations. With a gradient of a single morphogen POL, we may define a specified growth rate parallel to ∇POL (k_{par}), and a plane of directions perpendicular to this gradient. **Within that plane there can be an isotropic growth rate k_{per} (Figure 11).**

10. *Page 19: The authors write “this situation would correspond to specified growth rates being higher for internal compared to outer tissue layers”. If growth is a stress release process, the inside can release all of its residual stress, and still may not grow if it is constrained by the outer layer. In this sense growth itself does not put stress on the outer layer, turgor does, and it is more a matter of how much of the stress from turgor is borne by inner layers.*

We agree. See our response to point 2 above.

11. *Page 19: It would be interesting to see if the two scenarios (isotropic vs radial anisotropic growth) produce the same reaction to cutting experiments. Can't the cutting experiments be done in the model?*

We agree that it would be interesting. However, we have not yet implemented this, as we now remark on page 20:

Incisions could be implemented in GFtbox, and have been for sheet-like models, but we have not yet implemented this for volumetric models.

Some typos:

12. *Intro: “mathematically modelling” should be “mathematically model”.*

Changed.

13. *Figure 7: a,,b,c not referenced in caption.*

Now referenced.

Referee 2

General comments

1. ** Throughout the bulk of the present work, the authors rarely refer explicitly to biological applications. For example, when discussing anisotropic specified growth along one field of orientations, the authors refer to generic models where there are one or more polarity fields, but do not discuss concrete examples of such tissues. Including such references could help more closely tie the work to biology.*

We have added another model to illustrate a development that is beyond the capabilities of the previous version of GFtbox. This is illustrated in Figure 17, and described in the text on pp.18–20, in the last paragraph before the Discussion section.

To illustrate how this approach can be used to model processes that could not be captured using sheets alone, we modelled the formation of solid outgrowths from a leaf, observed in the *kanadi1 kanadi2* double mutant of *Arabidopsis* [36]. These outgrowths arise in leaf primordia at positions in the epidermis where planar polarity, as revealed by PIN1, converges. A subepidermal strand of PIN polarity also forms and runs through the centre of the outgrowth. To model this process, we modified the model used in Figure 16(h,i), by growing it to a primordial stage, and then introducing a –ORG for POL2 to create a site of planar polarity convergence (Figure 17(a)). We also created a strand of +ORG for POL, running from the convergence site towards the midplane (Figure 17(b)). Both k_{par2} and k_{per} were also increased in the neighbourhood of the +ORG strand (Figure 17(c,d)). With these assumptions an outgrowth formed, centred on the +ORG strand (Fig 17(e,f,g)). This model is beyond the capability of the earlier version of GFtbox, which was limited to sheet-like meshes which could vary in thickness, but did not support changes of form beyond that.

It is also briefly mentioned in the Discussion on p.21 and alluded to in the phrase “and outgrowths” at the foot of p.2.

2. ** On the other hand, the authors refer extensively to a previous publication (Kenaway et al, 2011) for technical details such as constitutive equations. It would be beneficial to have the constitutive equations at least briefly defined in the present manuscript - this would not take up much further space, and would save readers from having to look up another paper just to find out what equations are being solved in the present work.*

This has been added to the supplementary material (pp.1–2) in a new section “Constitutive equations”. (The diffusion equation was already present, but not the elasticity equation.)

3. ** From a computational perspective, what is the performance of the code? For example, do each of the simulations presented in Figures 6-16 take seconds, or minutes, or hours to run? Including a brief indication of this in the supplementary material would be beneficial to readers.*

We have added this to the Methods section of the supplement, p.1.

Specific comments

4. *p.3 “Dynamic Refinement” - The authors clearly describe dynamic remeshing of finite elements that are very large or thin. However, it wasn’t quite clear to me how this process is implemented across an entire mesh - do the authors randomise the order in which finite elements are refined after each time step? Does this order matter for the resulting overall mesh?*

Because of the possible interactions between simultaneous refinements of adjacent elements, we select from the candidates for refinement a maximal subset that can easily be transformed simultaneously. Candidates that were passed over can be split by repeated applications of this process. See the paragraph beginning “To ensure that we only deal with cases” on p.6.

5. *p.13 “Initial and final canvas” - Here and elsewhere in the manuscript, “canvas” seems to have a technical meaning, but I’m not aware what this is (maybe it is domain-specific); can the authors clarify?*

“Canvas” is a word that the second author introduced in other writings, to indicate modelled tissue that can grow while being patterned, and in which patterning can influence growth. For bulk solids rather than sheets it is perhaps a more obscure metaphor, and we have replaced the word throughout by “tissue”.

Typos

6. p.3 "*moprhogen*" → "*morphogen*"

p.8 "*with an initially circular sheet (Fig 7(a)).*" → "*with an initially circular sheet (Fig 7(a)).*"

p.25 "*movies of some the developments*" → "*movies of some of the developments*"

p.28 "*edges of tetrahedral mesh*" → "*edges of the tetrahedral mesh*"

p.29 "*vertexes*" → "*vertices*"

These have all been corrected (and several other occurrences of "*vertexes*").

References

- [1] Zhang T, Vavylonis D, Durachko DM, Cosgrove DJ. Nanoscale movements of cellulose microfibrils in primary cell walls. *Nature Plants*. 2017;3.
- [2] Kennaway R, Coen E, Green A, Bangham A. Generation of Diverse Biological Forms through Combinatorial Interactions between Tissue Polarity and Growth. *PLoS Comp Biol*. 2011;7(6):22pp.